# Photobiomodulation for Alzheimer's Disease: Has the Light Dawned?

**Michael R. Hamblin** [1,2,3]

[1]   Wellman Center for Photomedicine, Massachusetts General Hospital, Boston, MA 02114, USA;
      Hamblin@helix.mgh.harvard.edu
[2]   Department of Dermatology, Harvard Medical School, Boston, MA 02115, USA
[3]   Harvard-MIT Division of Health Sciences and Technology, Cambridge, MA 02139, USA

**Abstract:** Next to cancer, Alzheimer's disease (AD) and dementia is probably the most worrying health problem facing the Western world today. A large number of clinical trials have failed to show any benefit of the tested drugs in stabilizing or reversing the steady decline in cognitive function that is suffered by dementia patients. Although the pathological features of AD consisting of beta-amyloid plaques and tau tangles are well established, considerable debate exists concerning the genetic or lifestyle factors that predispose individuals to developing dementia. Photobiomodulation (PBM) describes the therapeutic use of red or near-infrared light to stimulate healing, relieve pain and inflammation, and prevent tissue from dying. In recent years PBM has been applied for a diverse range of brain disorders, frequently applied in a non-invasive manner by shining light on the head (transcranial PBM). The present review discusses the mechanisms of action of tPBM in the brain, and summarizes studies that have used tPBM to treat animal models of AD. The results of a limited number of clinical trials that have used tPBM to treat patients with AD and dementia are discussed.

**Keywords:** photobiomodulation; Alzheimer's disease; dementia; mechanisms of action; animal models; clinical trials

## 1. Introduction

### 1.1. Introduction to Photobiomodulation

Photobiomodulation (PBM) describes the therapeutic use of red or near-infrared light to stimulate healing, relieve pain and inflammation, and prevent tissue from dying. PBM used to be called "low-level laser (or light) therapy" (LLLT) but the name was changed to reflect the fact that the term "low" was undefined, lasers were not absolutely required, and inhibition of some processes was beneficial [1,2]. Photobiomodulation therapy (PBMT) describes the use of PBM as a treatment for various diseases or disorders. PBM was discovered over 50 years ago by Endre Mester in Hungary working with hair regrowth and wound healing in mice [3]. Since then, PBM has gradually become more accepted by the medical profession, physical therapists, and the general public. This increase in acceptance is partly due to the increased availability of light-emitting diodes (LEDs) with wavelengths in the red and NIR regions and substantial levels of power density (up to 100 mW/cm$^2$ over fairly large areas). Most available evidence suggests that LEDs perform equally well compared to lasers of similar wavelengths and power density [4]. However, LEDs have the advantages of more safety, lower cost, and better suitability for home use.

### 1.2. Mechanisms of PBM

It is the first law of photobiology that a photon must be absorbed by a specific molecular chromophore in order to have any biological effect. The chromophores that have been postulated to be

useful in PBM, absorb at different wavelength regions of the electromagnetic spectrum (blue, green, red, NIR), and are shown in Figure 1 and discussed below.

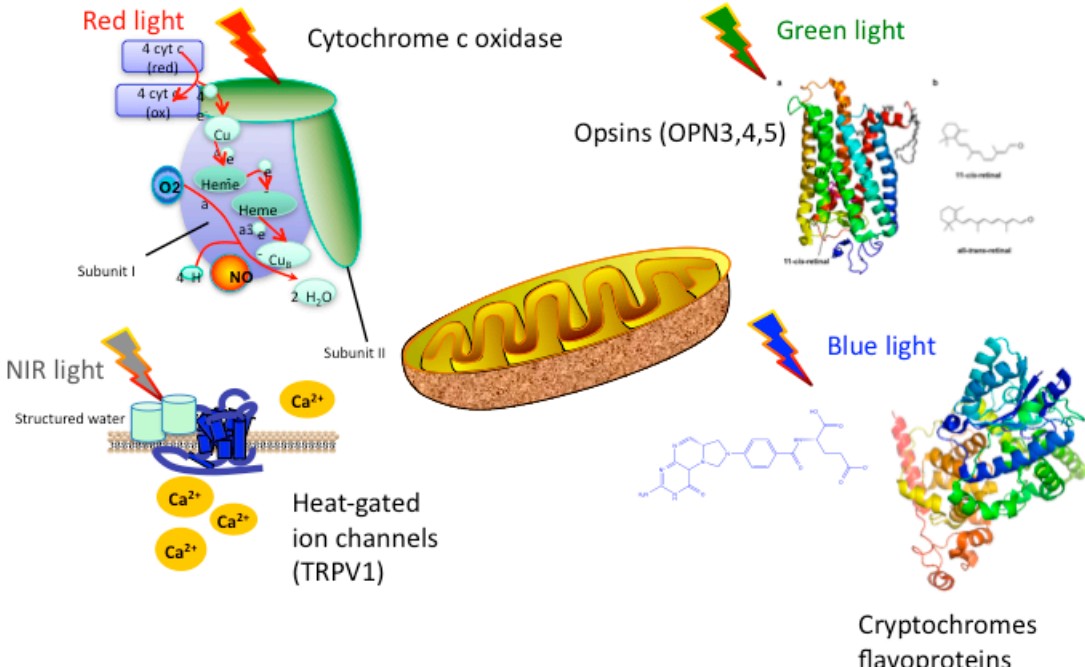

**Figure 1.** Proposed chromophores for PBM that can absorb different wavelengths of light. It should be noted that there is considerable overlap between the chromophores, and that the NIR absorbed by structured water is likely to be longer wavelength (>950 nm).

Cytochrome C oxidase (CCO) is the terminal enzyme (unit IV) in the electron transport chain situated in the outer mitochondrial membrane. The electron transport chain, through a series of redox reactions, facilitates the transfer of electrons across the inner membrane of the mitochondria. The net result of these electron transfer steps is to produce a proton gradient across the mitochondrial membrane that drives the activity of ATP synthase (sometimes called unit V) that produces the high-energy adenosine triphosphate (ATP) from ADP. CCO mediates the transfer of electrons from cytochrome C to molecular oxygen. CCO is a complex protein, composed of thirteen different polypeptide sub-units, and also contains two heme centers and two copper centers. Each of these heme and copper centers can be either oxidized or reduced, giving sixteen different oxidation states. Each of these oxidation states has a slightly different absorption spectrum, but CCO is almost unique amongst biological molecules in having a significant absorption in the near-infrared spectrum. In fact, Britton Chance estimated that over 50% of the absorption of NIR light by biological tissue could be attributed to this single enzyme as a chromophore [5].

In many publications, CCO has been shown to be a biological photoacceptor and transducer of signals activated by light in the red and NIR regions of the spectrum [6,7]. Specifically, absorption of the photons delivered in PBM, seems to promote an increase in the availability of electrons for the reduction of molecular oxygen in the catalytic center of CCO, increasing mitochondrial membrane potential (MMP), and increasing levels of ATP, cyclic adenosine monophosphate (cAMP), and reactive oxygen species (ROS), all of which indicate increased mitochondrial function, and can trigger initiation of cellular signaling pathways [8]. However recently, the CCO hypothesis has been brought into question. Lima et al. [9] genetically engineered two different kinds of cells to not express any active CCO, and found they responded equally well to 660 nm light, compared to wild type cells. Although other units in the electron transfer chain, such as complexes I-IV and succinate dehydrogenase also show increased activity as a result of PBM, CCO is still believed to be one of the primary photoacceptors.

This notion is supported by the fact that low-level light irradiation such as PBM causes increased oxygen consumption, and is bolstered by the fact that the majority of oxygen consumption occurs at complex IV, and moreover that the addition of sodium azide, a CCO inhibitor, abrogates the effects of PBM [10,11]. Moreover, rho-zero cells that lack functional mitochondria do not respond to PBM, in the same way as their wild-type counterparts [12].

Nevertheless, despite the amount of évidence in favor of CCO being a major chromophore for red and NIR light, mounting evidence is suggesting that this is not the whole story. Lima et al. [9] investigated two cell lines lacking CCO, one mouse line with the Cox10 knocked out (that could not synthesise the heme a cofactor) and a second human line with a mutation in the mtDNA gene coding for tRNA lysine (that lacked three critical CCO subunits). PBM (660 nm) caused increased cell proliferation in both wild type and CCO knock out cells, together with increased ATP and citrate synthase levels. These results showed that functional CCO was not required for its ability to enhance metabolism and cell proliferation.

A recent editorial [13] from Sommer in Ulm, Germany suggested that the effects of red and NIR light (especially pulsed at low frequency such as 1 Hz) on the interfacial water layers (IWL) inside cells could be an alternative explanation. If these IWL were inside the mitochondria, then the lowering of viscosity as a result of the energy absorption, could allow the molecular rotor, which is ATP synthase, to rotate faster and produce more ATP. On the other hand if the IWL were localized within the plasma membrane, light absorption could increase the uptake of nutrients accounting for increased proliferation.

Regardless of the actual chromophore, PBM can trigger retrograde mitochondrial signaling [14]. This refers to signals and communications passing from the mitochondria to the nucleus of a cell, rather than vice versa. The aforementioned mitochondrial changes result in an altered mitochondrial ultrastructure, and triggering of mitochondrial biogenesis [15]. As a result, membrane permeability and ion flux at the cell membrane are altered, in turn leading to the altered activity of activator protein-1 (AP1) and NFκB [16].

There is emerging evidence that other primary chromophores such as opsins, flavins and cryptochomes, may mediate the biological absorption of light, particularly at shorter wavelengths (blue and green). Opsins contain a *cis*-retinaldehyde molecule as a chromophore that is photo-isomerized to the all-trans isomer, thus producing a change in protein conformation and initiating a signaling cascade [17]. Flavins and flavoproteins contain a chromophore such as riboflavin, flavin mononucleotiode, or flavin adenine dinucleotide and can carry out redox reactions when excited by light [18]. Cryptochromes are a special sub-class of flavoproteins that act as blue–light receptors in plants, animals and even humans [19].

Although evidence proving that light-gated ion channels can be cited as mechanisms of action in PBM is sparse at the present time, it is gradually increasing. PBM is most likely to affect transient receptor potential (TRP) channels. First discovered in a Drosophila mutant as the mechanism responsible for the vision of insects, they are now known to be sensitive to light [20], in addition to a wide variety of other stimuli. TRP channels are calcium channels, and are modulated by phosphoinositides [21]. Light-gated ion channels have attracted immense attention in the field of optogenetics [22]. However the majority of these studies employ ion channels similar to bacterially derived channelrhodopsin [22]. The majority of research relating PBM to light-gated ion channels has been done by testing the TRPV "Vanilloid" subfamily of TRP species. Evidence from studies done by various groups [23–26] have led to the general consensus that TRP channels are most likely to be activated by green light. However, because green light lacks the same penetrating ability of infrared or near-infrared light, it lacks practical clinical application. However, Ryu et al. found that exposure to infrared (2780 nm) wavelength light attenuated TRPV1 activation, causing a decrease in generation of pain stimuli [24]. A similar, but far less dramatic antinociceptive effect was also observed when TRPV4 was exposed to light of the same wavelength. TRPV4 was also shown to be responsive to 1875 nm pulsed light, although it cannot be ruled out that

the results were due to thermal stimuli rather than light stimuli [25], as water is the primary absorber of infrared in this region.

It is clear that water must be by far the most important chromophore at infrared wavelengths (>900 nm), considering its molecular absorption coefficient and its relative abundance in cells and tissues. Nevertheless PBM as usually carried out, does not produce excessive heating of the tissues, especially within the brain. In fact the most noticeable heating effect (if any) is felt on the skin of the scalp. How then can we explain that PBM can have powerful effects on the brain at wavelengths as long as 1064 nm [27,28]? One answer may lie in the concept of 'nanostructured water' or 'interfacial water' elaborated by Pollack [29–31]. This exclusion zone (EZ) water (which may be the same as the IWL discussed above [13] absorbs optical radiation which produces distinct physical changes in parameters such as viscosity and pH. Since the EZ water layers occur on intracellular membranes, it is reasonable to suggest that ion channels embedded within these membranes (for instance in mitochondria), may be triggered by these physical changes. Since bulk water does not absorb IR light to the same degree as EZ water, this would explain why biochemical changes can take place within the cells, while there is no detectable bulk heating of the tissue, as would have been expected if the IR energy was absorbed by all water molecules.

## 2. Alzheimer's Disease and Dementia

Dementia is the clinical term used to describe a broad range of brain disorders that affect cognitive and executive functioning and memory [32]. The diagnosis of dementia requires a change in mental function with a more pronounced decline than one would expect due to the normal aging process [33]. In 2015, 46.8 million people throughout the world were estimated to be suffering from dementia, with 58% living in low and middle income countries and this number is expected to double every 20 years [34].

Alzheimer's disease (AD) is the most common type of dementia (60% to 70% of cases) followed by vascular dementia (25%), and Lewy body dementia (15%) [35]. AD was first described by Alois Alzheimer (1864–1915) who published his report in 1911 [36]. About 70% of the risk is probably genetic, with many genes proposed to be involved [37]. Other risk factors include a history of head injury, depression, and hypertension. AD is characterized by diffuse atrophy of the entire brain (especially of the cortex), accompanied by extracellular beta-amyloid plaques and intraneuronal neurofibrillary tangles composed of hyperphosphorylated tau protein [38]. The precise mechanisms of AD remain a subject of hot debate [39]. A wide variety of other investigational drugs have been tested in clinical trials, but so far without much success.

The following section will summarize some of the hypotheses. The amyloid hypothesis has been the predominant explanation for decades. Aβ peptides (40 or 42 amino acids) are formed by sequential enzymatic cleavage of amyloid precursor protein (APP) by beta and gamma secretases. An increase in the level of Aβ 42 leads to amyloid fibril formation, which eventually develop into senile plaques. However the failure of several drug trials that have targeted the amyloid peptides (beta and gamma secretase inhibitors) and amyloid plaques (immunotherapy approaches using monoclonal antibodies) has led to the concept that the amyloid plaques may be markers rather than causes of the brain deterioration [40].

An alternative hypothesis focuses on tau [41]. Tau is a microtubule-associated protein involved in microtubule assembly. There are two isoforms expressed in the adult human brain (4R and 3R) mainly in axons of neurons. In AD brains, 3R and 4R tau is accumulated in a hyperphosphorylated state that forms neurofibrillary tangles (NFTs) in cell bodies, or threads if they are formed in dendrites or axons. Many different brain disorders are characterized by tau pathology and are known as "tauopathies" [42]. These include frontotemporal dementia, corticobasal degeneration, Richardson syndrome, Parkinson's disease, chronic traumatic encephalopathy, and age-related tau astrogliopathy.

Neuroinflammation and reactive gliosis are hallmarks of AD [43]. Accumulating evidence suggests that that microglia with the M1 phenotype are important players in AD [44]. Not only do the M1 microglia pump out pro-inflammatory cytokines, but these cells down-regulate their phagocytic functionality, and therefore fail to clear the amyloid plaques. Any therapy (such as PBM) that can switch the microglial phenotype from M1 to M2 may be helpful for AD.

The increased incidence of AD in patients suffering from hypertension and irregular heartbeat, gave rise to the hypothesis of "micron strokes" [45]. Micro-strokes caused by fibrous eythrocyte emboli or micron-sized cholesterol crystals could act as "seeding points" for the growth of amyloid plaques as a healing response. A related hypothesis concerns the influence of vascular dysfunction and micro-hemorrhages [46]. Vascular dysfunction is often described as causing vascular dementia, but there is increasing evidence that it plays a role in AD as well [47]. These micro-hemorrhages have been correlated with plaque formation [48]. These micro-hemorrhages in cerebral vessels, could act as triggers to activate the innate immune system. They could also be indicative of sites of breakdown of the blood-brain barrier, which is considered as one of the early markers of cognitive dysfunction [49].

Oxidative stress has been implicated in the pathogenesis of AD [50]. The evidence includes increased levels of certain metals in AD brains such as iron, aluminum, and mercury that can generate free radicals. Increased lipid peroxidation, 4-hydroxynonenal, oxidative damage to protein and DNA, advanced glycation end products (AGE), malondialdehyde, carbonyls, peroxynitrite, heme oxygenase-1 and SOD-1 in neurofibrillary tangles and amyloid plaques. However although a diet high in antioxidants offers some protection, supplementation with antioxidants has largely failed to show any benefits [51].

Reductions in mitochondrial activity and glucose metabolism are widely seen in AD [52]. Changes in cytochrome c oxidase and morphological changes in mitochondria have been found. Activation of the integrated stress response and the transcription factor ATF4 may be caused by mitochondrial dysfunction.

Finally, another hypothesis implicates changes in the gut microbiome [53]. The bacteria themselves may secrete bacterial amyloid that may trigger cross-seeding of amyloid plaques, or else the bacteria may over-stimulate the innate immune response [54]. Bacteria themselves, such as *Porphyromonas gingivalis*, have been found in AD brains [55]. Other pathogens such as viruses and spirochetes may be involved in the brain, and Aβ peptide may function as an antimicrobial defense peptide [56].

## 3. Mechanisms of PBM in the Brain

As will be seen in the following section, a bewildering array of different mechanisms have been proposed to account for the benefits of transcranial PBM (tPBM) on the brain. These are schematically shown in Figure 2.

### 3.1. Metabolism

Improved metabolic functioning is one of the most easily recognizable effects of PBM, and increased intracellular ATP production is one the most strongly supported mechanisms of action [57]. Moreover, several pre-clinical studies have shown that the brain content of ATP was increased in experimental animals (mice or rats) subjected to tPBM for various brain disorders [58,59]. It is a general finding that mitochondrial dysfunction, inadequate supplies of ATP, and oxidative stress are contributory factors in almost all forms of brain disease [60]. This has been reported for neurological conditions such as major depressive disorder [61], traumatic brain injury [62], Parkinson's disease [63], and for AD [64].

### 3.2. Blood Flow

One of the changes that is easiest to measure after tPBM, is the change in cerebral blood flow and oxygenation. This applies in experimental animals, and especially in human subjects. Near-infrared spectroscopy has been used on the forearms of human volunteers treated with a 1064 nm laser [65]. They found that tPBM induced significant increases of CCO concentration (Delta [CCO]) and oxygenated

hemoglobin concentration (Delta [HbO]) in the treated site as the laser energy dose accumulated over time. Schiffer et al. [66] tested tPBM using an 810 nm LED applied to the forehead for major depression and anxiety, and used an INVOS commercial system from Somanetics (Troy, MI, USA) to measure cerebral hemoglobin (cHb) in left and right frontal and rCBF (regional cerebral blood flow), in addition to the device's usual oxygen saturation output.

**Figure 2.** There are a large number of mechanisms for tPBM in the brain that have been proposed (discussed below).

It has been suggested that the release of NO as result of PBM is responsible for the increased cerebral blood flow [67]. NO is a major neuronal signaling molecule which, among other functions, possesses the ability to trigger vasodilation. To do so, it first stimulates soluble guanylate cyclase to form cyclic-GMP (cGMP). The cGMP then activates protein kinase G, leading to the reuptake of $Ca^{2+}$ and the opening of calcium-activated potassium channels. Due to the subsequent fall in concentration of $Ca^{2+}$, myosin light-chain kinase is prevented from phosphorylating the myosin molecule, causing the smooth muscle cells in the lining of blood vessels and lymphatic vessels to become relaxed [68]. This vasodilation then promotes improved circulation, which in turn leads to improved cerebral oxygenation in a similar manner to that observed with pulsed electromagnetic fields [69].

Disorders of CBF, neurovascular dysfunction, and lower brain oxygenation have been proposed to an important feature of AD [70].

*3.3. Neuroprotection*

A wide variety of evidence suggests that PBM can be utilized for neuroprotection, essentially, to protect cells from damage, to promote their survival and longevity, and reverse apoptotic signaling processes. One way it achieves this result is by inhibiting the activity of glycogen synthase kinase 3β (GSK3β). To do so, it activates protein kinase B (AKT), which increases the phosphorylation level

of its Ser9 residue, which then allows the N -terminus of GSK3β to bind with its own binding site. One result of this is the accumulation and translocation to the nucleus of β-catenin, which ceases to be under-phosphorylated and therefore becomes more active when GSK3β activity is inhibited. Once allowed to accumulate in the nucleus, β-catenin relies on the increased TCF/LEF dependent transcriptional activity to promote cellular survival [71]. This inhibition of GSK3β also helps to prevent apoptosis, the normal cell death that occurs as an organism grows. GSK3β is believed to act as a mediator between AKT and Bax, a protein which is translocated to the nucleus in the presence of pro-apoptotic stimuli to trigger the beginning of the process. However, when GSK3β is inhibited, AKT the communication pathway between AKT and Bax is cut off. As a result, Bax translocation cannot be signaled for and is thus inhibited [72].

PBM also demonstrates neuroprotective qualities in the form of protection from senescence [73]. It has been shown to activate the extracellular signal-related kinase (ERK)/forkhead box protein M1 (FOXM1) pathway. The FOXM1 protein regulates the progression from the G1 to the S phase of the cell cycle, and via the activation of the ERK/FOXM1 pathway, PBM leads to the greater translocation of ERK to the nucleus and the greater accumulation of FOXM1 in the nucleus. This, in turn, causes reduced expression of the p21 protein and mitotic arrest in the G1 phase, therefore slowing the overall progression of cellular senescence.

PBM has also been shown to be effective in protecting cells from the harmful effects of toxins [74]. In a study done by Eells et al. [75], irradiation with 670 nm light was successful in causing the recovery of retina function and the prevention of histological damage in rodent models exposed to methanol. This is likely due to the fact that methanol generates the toxic metabolite formic acid, an inhibitor of CCO, and PBM is a known stimulator of CCO. A study by Wong-Riley on the effects of PBM post-tetrodotoxin exposure produced similarly successful results, especially when models were irradiated with 670 and 830 nm light, the peaks of the CCO absorption spectrum [7]. This further indicates that the antitoxin effect of PBM can be traced to its stimulation of CCO. PBM is also effective in prevention of the harmful effects associated with potassium cyanide. When pretreated with 670 nm light, Liang et al. in 2006 found that neuronal expression of Bax induced by cyanide was decreased, preventing the subsequent apoptosis [76].

In addition, PBM has demonstrated the rather unique property of affecting cells in different states of health in different ways, essentially modifying the cell in whatever way might be necessary to promote its survival. For instance, in normal cells the absorption of light by CCO leads to an increase in MMP above baseline and a short surge in ROS production. However, in cells where MMP is low due to existing oxidative stress, excitotoxicity, or inhibition of electron transport, light absorption leads to an increase of MMP towards normal levels and a decrease of ROS production [77]. Similarly, the typical response to PBM in healthy cells is a brief increase in intracellular $Ca^{2+}$ [78]. However, in cells that already contain excess $Ca^{2+}$ (a phenomenon called excitotoxicity) PBM provokes the opposite reaction, in other words it lowers excessive levels of cellular calcium, thus promoting cell survival, lowering oxidative stress, and raising MMP back to normal [79]. A range of neuroprotective approaches based on natural products are under investigation for treatment of AD [80].

### 3.4. Oxidative Stress

Oxidative stress occurs when there exists an imbalance between the production of reactive oxygen species (ROS) and the ability of the body to counteract their effects, which become harmful when they are in excess, via antioxidants. Many sources have linked oxidative stress to various neurological conditions, such as major depressive disorder [81] and traumatic brain injury [82], not to mention cardiovascular [83] and Alzheimer's diseases [84].

However, the situation is more complicated than at first appears, because large numbers of clinical trials of antioxidant therapy for all these diseases, have failed (sometimes dismally) [51,85]. Apparently some level of oxidative stress is necessary for the optimum functioning of human beings, and removing all oxidative stress with supplementation with anti-oxidants can be counter-

productive [86]. An important paper showed that the health giving benefits of exercise were removed when humans were administered antioxidants [87].

Salehpour et al. [88] showed that sleep deprivation (SD) in mice caused oxidative stress in the hippocampus and subsequent memory impairment. tPBM with NIR (810 nm) was delivered (once a day for 3 days) transcranially to the head. Mice performed better on the Barnes maze and the What-Where-Which task, and hippocampal levels of antioxidant enzymes were increased and oxidative stress biomarkers were decreased. In studies of the effect of PBM on traumatized muscle, PBM has been shown to be effective in regulating the amount of cytokine-inducible nitric oxide synthase (iNOS) produced by the cell. This is important because excessive amounts of iNOS can lead to the excessive production of NO, which would then signal increased production of the ROS/RNS called peroxynitrite, leading to an increase in oxidative stress. Specifically, PBM could reduce peroxynitrite [89], while still preserving the positive effects of other isoforms of NO synthase, such as endothelial nitric oxygen synthase (eNOS), which is the species primarily responsible for the vasodilating effects of PBM [90–92].

PBM has also been shown to stimulate increases in angiogenesis, leading to further improvements in blood flow. As demonstrated by Cury [93], PBM at 780 nm and 40 J/cm$^2$ triggered an increase in the expression of the protein HIF 1$\alpha$ and of vascular endothelial growth factor, and a decrease in matrix metalloproteinase 2 activity, all of which were found to induce angiogenesis. Additionally, in an in vitro study of the effects of red/NIR light on red blood cells, NIR light was found to be quite effective in protecting red blood cells from oxidation [94], which is a common occurrence in brains compromised by conditions such as MDD [95].

The widespread mitochondrial dysfunction, increased levels of aluminum and heavy metals, and neuroinflammation that occur in AD, produce significant levels of oxidative stress. Oxidative stress causes A$\beta$ deposition, tau hyperphosphorylation, and the subsequent loss of synapses and neurons [96]. A variety of antioxidants (and in particular coenzyme Q10) have been tested for treatment of AD [97].

### 3.5. Anti-Inflammatory Effects

Inflammation is the one of the innate immune system's defenses against foreign bodies such as bacteria and viruses. On a cellular level, it occurs when the transcription factor NF -$\kappa$B is activated. While acute inflammation is positive, chronic inflammation can have very negative effects. Many diseases, including neurodegenerative diseases and mood disorders, can be traced at least in part to chronic inflammation.

One way PBM helps to quell inflammation is through the inhibition of the cyclo-oxygenase 2 (COX-2) enzyme. Lim et al. found [98] that 635 nm light irradiation at low power was able to cause COX-2 inhibition by decreasing intracellular ROS. Inhibition of COX-2 via pharmaceutical means is widely supported at present, with COX-2 inhibitors making up a significant portion of the market for non-steroidal anti-inflammatory drugs (NSAIDS [99]). Using PBM, essentially the same result can be accomplished, just with a different stimulus.

PBM can also modulate cellular levels of free NF $\kappa$B. NF $\kappa$B is found in the cytosol bound to I$\kappa$B, an inhibitor protein. Pro-inflammatory stimuli have the ability to activate I$\kappa$B kinase, an upstream signaling regulator that causes the degradation of I$\kappa$B. Once the I$\kappa$B has been degraded, the NF $\kappa$B is free to translocate to the nucleus, where it triggers the expression of pro-inflammatory genes. There is evidence that PBM can have opposite effects on NF $\kappa$B depending on the type of cells and their activation state that is studied. Chen et al. found that in normal fibroblasts PBM could activate NF $\kappa$B via generation of low amounts of ROS from mitochondria that had been stimulated [100]. The same group however, found that in dendritic cells (another type of macrophage cell) that had been activated towards a M1 phenotype by toll like receptor agonists, that PBM could reduce pro-inflammatory cytokines [101]. Likewise, Yamaura et al. found that the level of NF-kB was reduced in activated rheumatoid arthritis-derived synoviocytes that received PBM [102].

Additionally, PBM possesses the ability to modulate levels of cytokines, proteins that act as important signaling molecules for the immune system. PBM has been shown to modulate levels of both

pro and anti- inflammatory cytokines, though for the reduction of inflammation, its ability to modulate levels of tumor necrosis factor (TNF) and other pro-inflammatory cytokines is especially useful.

It should be noted that inflammation within the brain has distinct differences compared to inflammation in other parts of the body. In fact, the term 'neuroinflammation' is commonly applied to the activation of microglia. Microglia are cells of the monocyte/macrophage lineage that act as the immune defense system in the central nervous system [103]. Microglia are constantly scavenging the CNS for plaques, damaged neurons and synapses, and infectious agents. Microglia are extremely sensitive to even small pathological changes in the CNS [104].

In common with other cells in the macrophage lineage, microglia can assume a diversity of phenotypes, and retain the capability to shift their function to maintain tissue homeostasis. Microglia can be activated by LPS or IFN-γ to an M1 phenotype that expresses pro-inflammatory cytokines and is able to kill microbial cells. On the other hand microglia can be activated by IL-4/IL-13 to an M2 phenotype for phagocytosis of debris, resolution of inflammation and tissue repair. Increasing evidence suggests a role of metabolic reprogramming in the regulation of the innate inflammatory response [105]. Studies have demonstrated that the M1 phenotype is often accompanied by a shift from oxidative phosphorylation to aerobic glycolysis for energy production [106]. Under these conditions, energy demands would be associated with functional activities and cell survival and thus, may serve to influence the contribution of microglia activation to various neurodegenerative conditions.

Since there is considerable evidence that PBM can activate the mitochondrial metabolism towards oxidative phosphorylation, and away from aerobic glycolysis this is a plausible reason why PBM may change the microglial phenotype from M1 towards M2 [107]. The consequences of this shift would be that instead of M1 microglia that cannot dispose of substances such as beta-amyloid plaques in AD, and therefore generate ROS and inflammatory cytokines, PBM-induced M2 microglia could clear the plaques, exert anti-inflammatory and anti-oxidant effects and encourage tissue healing [108].

*3.6. Neurogenesis*

For many years, it was thought that the adult brain was incapable of growing new brain cells. Although it was realized that growing and developing brains in embryos, young animals and children must be capable of neurogenesis mediated by neural stem cells (NSCs) and neuroprogenitor cells, it was though that this process had ceased in adulthood. The turning point in our perception was the discovery of adult neurogenesis and identification of cells that both in vitro and in vivo can function as NSCs, generating new neurons, glial cells, or both [109]. The paradigm shift regarding the nature of NSCs and the potential of the post-natal brain to regenerate opened the gates for new studies with a new outlook [110]. Now the scientific community is engaged in not only in depth understanding of the adult brain neurogenesis and NSC functions but also how they may be encouraged with novel treatment modalities [111]. Experimentally NSCs/NPs are detected by the incorporation of bromodeoxyuridine (BrdU) into the nuclei of dividing cells after infection into the animal at various times before sacrifice, that can be subsequently measured by an antibody [112]. However it has been established that there are only a few well-defined areas of the brain in which this neurogenesis is observed which are known as "neurogenic niches" [113]. The most well accepted neurogenic niches are the sub-granular layer of the dentate gyrus of the hippocampus [114], and the subventricular zone (SVZ) of the lateral ventricles [115]. In order to be assured the BrdU positive cells are actually neurons, rather than glia or some other cell type, it is usual to stain them with a second antibody to NeuN (marker of mature neurons) or to Tuj-1 (beta tubulin class III) [116].

The first report of neurogenesis being stimulated by tPBM delivered to the brain came from a study by Oron et al. in 2006, who induced a stroke in rats, and treated them with tPBM. The number of newly formed neuronal cells (BrdU-Tuj-1 double-positive) as well as migrating cells (doublecortin positive), was significantly elevated in the subventricular zone of the hemisphere ipsilateral to the induction of stroke when treated with PBM [117]. A similar result was reported by Xuan et al., who treated mice that had suffered a TBI using tPBM [118]. They found that there was a significant increase in

double-stained BrdU-NeuN (neuroprogenitor cells) in the dentate gyrus and in the SVZ at 1-week post TBI but not at 4-weeks post-TBI. Increases in double-cortin (DCX) and TUJ-1 were also seen. A recent report [119] showed that there was a sharp drop in hippocampal neurogenesis in subjects with AD, and this reduction increased along with disease progession.

### 3.7. Synaptogenesis

One of the most notable and potentially significant effects of tPBM on the brain discovered to date, is its ability to promote synaptogenesis, also called neuroplasticity. This process is vitally important, as many brain conditions, including TBI, stroke, neurodegenerative diseases, and mood disorders can be traced, either partially or in full, to poor or aberrant neuronal connections in certain regions of the brain. If tPBM possesses the ability to counter these effects by facilitating neural organization or reorganization, it could prove to be extremely promising as a novel method of treating such brain disorders.

One manner in which tPBM promotes neuronal connectivity could be by up-regulation of BDNF (brain derived neurotrophic factor). It is a member of the class of 'neurotrophins' which also includes nerve growth factor (NGF), NT3, NT4 and GDNF [120]. BDNF is a protein found in the nervous system, which helps to maintain existing neurons and to encourage the growth of new neurons and new synapses. Specifically, it is believed to modulate dendritic structure to facilitate improved synaptic transmission [121]. PBM has been shown to slow attenuation of BDNF via the ERK/CREB pathway, thus positively affecting dendritic morphogenesis and improved neuronal connectivity [122]. BDNF is also a mediator of the protein synapsin-1, which improves synaptogenesis by accelerating the development of neuronal fibers and maintaining synaptic contacts [123]. In a study carried out by Meng et al [124], denser branches and increased interconnectivity between fibers were observed in neural tissue of embryonic rats following irradiation with 780 nm light, indicating increased activity of these proteins. BDNF has also been linked to improvements in neuroplastic adaptation, which is especially important in cases of traumatic brain injury and stroke [125].

If it can be conclusively shown that tPBM stimulates neuroplasticity and synaptogenesis in humans as well as mice, then this opens the door to a wide range of clinical applications [126]. Impaired or aberrant neuroplasticity has been implicated in a wide range of brain disorders such as Alzheimer's [127], psychiatric disorders [128], stroke [129], TBI [130], and addiction [131].

### 3.8. Stem Cells

It should not be forgotten that when any kind of PBM light is shone onto a living animal, it is inevitable that some stem cells will be exposed to light. It is known that stem cells respond well to PBM in terms of proliferation and differentiation [132,133]. The stem cells may be located in the bone marrow underlying the tissue, and in the bones, which are in the illuminated area. Oron et al. [134] showed that applying PBM to the bone marrow in the legs had a therapeutic effect in a mouse model of Alzheimer's disease. The same procedure had major therapeutic benefits for reducing the infarct area in heart attack models [135,136], and in ameliorating ischemic kidney injury [137]. Recently, clinical trials using mesenchymal stem cells and neural stem cells have been carried out for AD, although as yet no efficacy has been observed [138].

### 3.9. Gamma Rhythms

In 2016 an important study from Iaccarino et al., working at MIT, reported that 40 Hz pulsed blue light could reduce the load of amyloid-β (Aβ)1-40 and Aβ 1-42 peptides and lessen the amyloid plaque burden in the visual cortex of the brain in a mouse model of AD [139]. This effect was originally discovered using an optogenetic technique to stimulate fast-spiking parvalbumin-positive (FS-PV)-interneurons. The hypothesis was that the microglia in the brain were transformed to an "engulfing state" by the gamma entrainment. A subsequent study from the same group showed that 40 Hz pulsed light was neuroprotective, and improved cognitive performance in the Tau P301S

and CK-p25 mouse models of neurodegeneration [140]. They found improved synaptic function, enhanced neuroprotective factors, and reduced DNA damage in neurons, while the inflammatory response in microglia was reduced. This group went on to test a combination of 40 Hz blue light with 40 Hz auditory stimulation to produce "gamma entrainment using sensory stimulus" (GENUS) [141]. This combined approach produced reduction of amyloid plaques in much wider areas of the mouse brain and improved cognitive function.

## 4. tPBM for AD in Animal Models

One of the most convincing reports of the benefits of PBM in animal models of AD was carried out by De Taboada et al. in 2011 [142]. They delivered 810 nm laser (3×/week for 6 months) to the heads of amyloid-β protein precursor (AβPP) transgenic mice. The numbers of Aβ plaques were significantly reduced in the brain by administration of PBM in a dose-dependent manner. Administration of PBM produced a dose-dependent reduction in amyloid load, soluble AβPPα, and brain inflammatory markers. ATP levels, mitochondrial function, and c-fos were all increased. Cognitive function as measured by the Morris water maze was improved by PBM.

Purushothuman et al. [143] reported the beneficial effects of PBM in two separate mouse models of AD, each designed to display relevant pathological changes in the brain. These were the K369I tau transgenic model (engineered to develop neurofibrillary tangles), and the APPswe/PSEN1dE9 transgenic model (engineered to develop amyloid plaques). Mice were treated for 90 s with 670 nm LEDs 5×/week for 4 weeks. In the tau mice, tPBM produced a reduction in hyperphosphorylated tau, neurofibrillary tangles and oxidative stress markers (4-hydroxynonenal and 8-hydroxy-2'-deoxyguanosine) to near control levels in the neocortex and hippocampus, and restored expression of mitochondrial cytochrome c oxidase in surviving neurons. In the amyloid-β mice, PBM reduced the size and number of amyloid-β plaques in the neocortex and hippocampus. A follow up report from the same group extended these observations to the cerebellum region of the mouse brain [144].

Farfara et al. [134] used a different transgenic mouse model, 5XFAD transgenic male mice (Tg6799) that co-overexpress familial AD (FAD) mutant forms of human APP (the Swedish mutation, K670N/M671L; the Florida mutation, I716V; and the London mutation, V717I) and PS1 (M146L/L286V) trans-genes, under transcriptional control of the neuron-specific mouse Thy-1promoter [145]. Moreover, these investigators treated the bone marrow in the mouse leg, instead of the head. An 810 nm laser connected to a fiber optic cable was used in contact with the tibia after making a small incision in the skin to deliver 1 J/cm$^2$. Mice were treated with PBM six times (at 10-day intervals, for 2 months) starting at the age of 4 months. Treated mice showed improved cognitive performance as measured by the object recognition test and the fear conditioning test. Immunohistochemical analysis of brain slices showed the PBM-treated mice had a 68% reduction in amyloid plaque burden.

Some investigators have taken a different approach to developing rodent models of AD, by directly injecting different kinds of Aβ peptides into the hippocampus. Lu et al. [146] used rats injected with Aβ 1-42 and applied 808 nm light to the head for 2 min/day for 5 days. tPBM ameliorated the neurodegeneration in the hippocampus and improved long-term spatial and recognition memory. Molecular studies showed that PBM: (1) improved mitochondrial dynamics; (2) raised mitochondrial membrane potential; (3) reduced oxidized mitochondrial DNA and mitophagy; (4) inhibited apoptosis as shown by the Bcl-2-associated X protein/B-cell lymphoma 2 ratio; (5) increased mitochondrial antioxidant expression; (6) raised cytochrome c oxidase activity and ATP levels; (7) suppressed Aβ-induced reactive gliosis, inflammation, and tau hyperphosphorylation.

da Luz Eltchechem et al. [147] also used rats but injected the hippocampus with Aβ25-35 peptide and treated them daily with 627 nm laser for 100 s, 7 J/cm$^2$ for 21 days. The use of tPBM significantly reduced the Aβ plaques, and improved spatial memory and behavioral and motor skills in treated animals on day 21. Blivet et al. [148] used male Swiss mice and also injected Aβ 25-35 peptide into the hippocampus. tPBM used a RGn500 device that emitted 850 nm laser and LED and 625 nm LED all pulsed at 10 Hz and surrounded with a ring-shaped magnet creating a 200 mT static magnetic field.

The device delivered 8.4 J/cm$^2$ over 10 min, and was applied once a day for 7 days either on the top of the head or the center of the abdomen or both. Protection from neurotoxicity was seen whether the PBM was applied to the head and the abdomen together, but not to either alone. Mice showed improved memory (by Y maze and passive avoidance test), while Aβ 1-42, pTau, oxidative stress (lipid peroxidation), apoptosis (Bax/Bcl2) and neuroinflammation were all reduced.

## 5. Clinical Trials of PBM in AD and Dementia

One of the first human trials of tPBM for AD to be reported, was reported by Saltmarche et al. [149] who treated a case series of five patients diagnosed with mild to moderately severe dementia (Mini-Mental State Exam, MMSE, scores of 10–24). The study used the Vielight alpha (810 nm, 10 Hz pulsed, LEDs) that combines transcranial plus intranasal PBM to treat the cortical nodes of the default mode network (bilateral mesial prefrontal cortex, precuneus/posterior cingulate cortex, angular gyrus, and hippocampus). There was 12 weeks of active treatment consisting of weekly in office transcranial/intranasal, and daily home-based intranasal, plus a 4-week no-treatment follow up. At 12 weeks, cognitive function was significantly increased (MMSE and ADAS-cog), sleep was improved, fewer angry outbursts, less anxiety, and wandering were reported post-PBM. There were no negative side effects. Precipitous declines were observed during the 4-week follow-up no-treatment period.

Recently Chao published [150] a second pilot trial testing the effects of home PBM on cognitive and behavioral function, cerebral perfusion, and resting-state functional connectivity in eight patients (four tPBM and four usual care) diagnosed with dementia. The PBM treatments were administered at home three times per week with the Vielight Neuro Gamma device. The participants were assessed with the Alzheimer's Disease Assessment Scale-cognitive (ADAS-cog) subscale and the Neuropsychiatric Inventory (NPI) at baseline and 6 and 12 weeks, and with arterial spin-labeled perfusion magnetic resonance imaging (MRI) and resting-state functional MRI at baseline and 12 weeks. After 12 weeks, there were improvements in ADAS-cog (p = 0.007) and NPI (p = 0.03), increased cerebral perfusion (p < 0.03), and increased connectivity between the posterior cingulate cortex and lateral parietal nodes within the default-mode network in the tPBM group.

Berman et al. [151] carried out a small pilot double blind, placebo-controlled trial in subjects diagnosed with dementia (n = 11, including six active, three controls and two dropouts) to assess the effect of 28 consecutive, six-minute transcranial sessions of NIR PBM using 1060–1080 nm LEDs embedded in a helmet. The results showed improvement in executive function, clock drawing, immediate recall, praxis memory, visual attention and task switching (Trails A&B) as well as a trend for improved EEG amplitude and connectivity measures.

Salehpour et al. [152] reported the treatment of a single case, who had been diagnosed with cognitive decline and olfactory dysfunction. The patient received twice-daily PBM therapy at home using three different wearable LED devices. For the first week a prototype transcranial light helmet and a body pad were used containing a mixture of CW 635 nm and 810 nm LEDs. The body pad was placed on various areas on the lower back and the helmet was worn while seated. After the first week of treatment, an intranasal LED device, 10-Hz 810 nm, was initiated in the left nostril twice daily. All three devices were applied simultaneously for an irradiation time of 25 min per session. The patient showed a significant improvement in the Montreal Cognitive Assessment score from 18 to 24 and in the Working Memory Questionnaire score from 53 to 10. The olfactory dysfunction was reversed as measured by the Alberta Smell Test and peanut butter odor detection test. Quality-of-life measures improved and caregiver stress was reduced.

Ivan Maksimovich working in Moscow, Russia has treated a large number of patients with AD using an intravascular catheter approach to deliver 632-nm light (25 mW HeNe laser) into the brain [153]. Under local anesthesia, the common femoral artery was catheterized and a thin, flexible, fiber-optic (25 to 100 μm) was advanced to the distal sections of the anterior and middle cerebral arteries where PBM was performed taking 20–40 min [154]. 93 patients aged 34–80 (32 men, 61 women) with

AD severity stages, TDR-0 (preclinical stage)—10 patients; TDR-1 (early stage)—26 patients; TDR-2 (middle stage)—40 patients; TDR-3 (late stage)—17 patients. Of these 48 patients received transcatheter intracerebral PBM, while 45 patients in the control group received conservative treatment with memantine and rivastigmine. PBM treated patients showed improvement of cerebral microcirculation, reduction of dementia and restoration of cognitive functions. The control group did not show and significant changes. The mechanisms were proposed to involve increased capillary blood supply, improved tissue metabolism, stimulation of neurogenesis, and the clearance of amyloid beta.

## 6. Devices and Parameters for Brain tPBM

There exists a wide range of devices and parameters that have been used for PBMT for AD in humans. Both lasers and LEDs have historically been used on the head, but as time goes on, LED arrays are becoming increasingly the most popular method of delivering PBM to the head. NIR wavelengths in the 800–900 nm range are the most popular choice, but 1064 nm or 1080 nm has also been used. Some investigators combine a red wavelength such as 660 nm with the NIR. The precise placement of the LEDs on the head also varies. Since the forehead is without hair, which can block the light, it is a popular choice for tPBM. Some helmets and hoods are designed to deliver light covering the entire head. Figure 3 shows a selection of devices that have been used for tPBM.

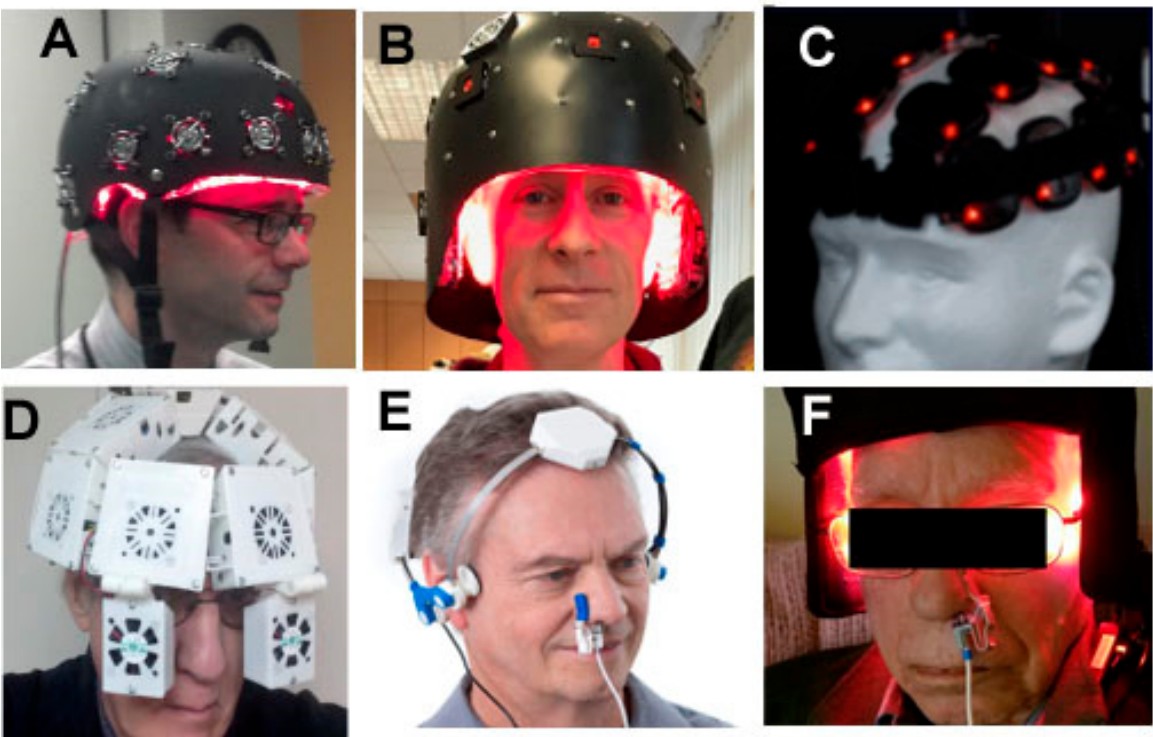

**Figure 3.** Selection of tPBM devices that have been clinically tested in AD and other brain disorders. (**A**) Helmet from Photomedex Inc. (Philadelphia, PA, USA); (**B**) Helmet from THOR Photomedicine (Chesham, UK); (**C**) Lumiwave LED clusters from BioCare Systems (Parker, CO, USA); (**D**) Helmet from Cognitolite (Dublin, Ireland); (**E**) Neuro-alpha LED device from Vielight (Toronto, ON, Canada); (**F**) Device from ProNeuroLIGHT LLC (Phoenix, AZ, USA).

The dosimetry for tPBM is measured as energy density in $J/cm^2$ and in total energy (J). Values of energy density range from 10–60 $J/cm^2$, but in my opinion the most useful measure of dose is total energy in Joules, which can be as large as several thousand J when fairly large arrays are used. For instance if a 500 $cm^2$ area of the head is exposed for 10 min to 20 $mW/cm^2$ then a total dose of 6000 J is delivered. Whole body light beds will deliver hundreds of thousands of J, but as yet there

is no research on this method of PBM for AD or brain disorders. Pulsing of the light is increasingly being focused upon, particularly at 10 Hz to stimulate alpha rhythms or 40 Hz to stimulate gamma rhythms. Intranasal light delivery is often used either on its own or to boost tPBM, but here the doses are much lower since typically a single 10 mW LED is clipped into the nostril. Finally the intravascular approach of Maksimovich must be remembered [153,154]. The preliminary report from Oron who obtained good results in a mouse AD model by applying PBM to stimulate autologous stem cells in the bone marrow of the tibia [134], suggests that this approach may be worth trying in humans.

As yet there have been no studies on combination therapies including PBM for AD. However preliminary animal studies suggest that tPBM would work well when combined with coenzyme Q10 [155–157]. There has been considerable interest in the use of coenzyme Q10 supplements in AD and other neurodegenerative diseases [158].

## 7. Future Perspectives

The fact that PBMT may produce a large range of beneficial changes in the brain, and is without any major side-effects, suggests it should be more widely tested for AD and dementia in large controlled trials. Exposing the head to light at power levels less than that received in direct sunlight (but without harmful ultraviolet wavelengths) is intrinsically safe. Any side-effects reported have been rare, mild and transient, consisting of slight headache, difficult sleeping and mild itching on the scalp. It is likely that tPBM for AD will need to be continued indefinitely, as regressions have been observed when PBM treatments have ceased. Moreover, unrelated health problems such as urinary tract infections or falls can lead to loss of the benefits achieved with tPBM. Home use tPBM devices can be applied by the caregivers, who consistently report improvements in their own quality of life.

**Author Contributions:** M.R.H. conceived and wrote the entire article.

**Funding:** M.R.H. was supported by US NIH Grants R01AI050875 and R21AI121700.

**Conflicts of Interest:** M.R.H. declares the following potential conflicts of interest. Scientific Advisory Boards: Transdermal Cap Inc., Cleveland, OH, USA; BeWell Global Inc., Wan Chai, Hong Kong; Hologenix Inc. Santa Monica, CA, USA; LumiThera Inc., Poulsbo, WA, USA; Vielight, Toronto, ON, Canada; Bright Photomedicine, Sao Paulo, Brazil; Quantum Dynamics LLC, Cambridge, MA, USA; Global Photon Inc., Bee Cave, TX, USA; Medical Coherence, Boston, MA, USA; NeuroThera, Newark, DE, USA; JOOVV Inc., Minneapolis-St. Paul, MN, USA; AIRx Medical, Pleasanton, CA, USA; FIR Industries, Inc. Ramsey, NJ, USA; UVLRx Therapeutics, Oldsmar, FL, USA; Ultralux UV Inc, Lansing, MI, USA; Illumiheal & Petthera, Shoreline, WA, USA; MB Lasertherapy, Houston, TX, USA; ARRC LED, San Clemente, CA, USA; Varuna Biomedical Corp. Incline Village, NV, USA; Niraxx Light Therapeutics, Inc., Boston, MA, USA. Consulting; Lexington Int., Boca Raton, FL, USA; USHIO Corp., Japan; Merck KGaA, Darmstadt, Germany; Philips Electronics Nederland B.V. Eindhoven, Netherlands; Johnson & Johnson Inc., Philadelphia, PA, USA; Sanofi-Aventis Deutschland GmbH, Frankfurt am Main, Germany. Stockholdings: Global Photon Inc., Bee Cave, TX, USA; Mitonix, Newark, DE, USA.

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
