# Peer review of "Photobiomodulation for Alzheimer’s Disease: Has the Light Dawned?"

_photonics, doi:10.3390/photonics6030077_

Round 1
Reviewer 1 Report
This manuscript provides an excellent overview of the present state of understanding of Alzheimer's disease, a multifactorial and complex neurodegenerative condition and the possible roles photobiomodulation may play in ameliorating some symptoms of the disease. The well-organised script is logically building up the arguments for the use of PBM. Importantly, these arguments are based on testable hypotheses and are illustrated by results from preclinical studies. The few, small clinical studies seem to support the notion that PBM may be a valuable adjunct therapy. As PBM is non-invasive therapy with no or very mild side-effects, it can be well tolerated by the patients and used by health professionals and care givers.
Major comments:
The disease mechanisms of AD and dementia listed in section 2, are well aligned with the hypotheses on PBM (e.g. oxidative stress, neuroinflammation...). One hypothesis should also be mentioned, which is related to vascular dysfunction and microhaemorrhages. Vascular dysfunction is often described as causing vascular dementia, but there is increasing evidence that it plays a role in AD as well (Sweeney at al, 2019, Alzheimer's & Dementia 15:158-167). Although 'micron strokes' are mentioned, earlier work already suggested of the presence of micro haemorrhages and their correlation with plaque formation (Cullen at al, 2005, J Cer Blood Flow 25:1656-1667; Cullen et al 2006; Neurobiol Aging 27:1786-1796) . As these microhaemorrhages, are closely related to cerebral vessels, they may present triggers to activate the innate immune system, but can also indicate sites of BBB breakdown, which is considered as one of the early markers of cognitive dysfunction (Nation et al. 2019 Nat Med 25:270-276). The addition of these would enrich this Section and would avoid any perceived bias towards hypotheses that directly support PBMt, without the loss of a valid argument for the use of PBMt in conditions with early vascular dysfunction.
Minor comments:
tPBM and PBMT are not defined. PBMT only appears on the last 2 pages and it could be used interchangeably in the text, however, I assume they indicate 2 separate concepts (tPBM=transcranial PBM? PBMT=photobiomodulation treatment?) Please clarify
P3, Lines 101 & 106 Cryptochromes are misprint
P3 L105 cary misprint
P7 L 274 'uptick'?
P8 L295. 'for' should be 'were'
P9 L372 '...it was thought...' (t is missing)
P11 L470 'deliver' misprint
P12 L478 'neurodegeneration' misprint
P12 L507 'the effects of' is repeated 2x
P19 Reference #118 is invalid citation
Author Response
Major comments:
The disease mechanisms of AD and dementia listed in section 2, are well aligned with the hypotheses on PBM (e.g. oxidative stress, neuroinflammation...). One hypothesis should also be mentioned, which is related to vascular dysfunction and microhaemorrhages. Vascular dysfunction is often described as causing vascular dementia, but there is increasing evidence that it plays a role in AD as well (Sweeney at al, 2019, Alzheimer's & Dementia 15:158-167). Although 'micron strokes' are mentioned, earlier work already suggested of the presence of micro haemorrhages and their correlation with plaque formation (Cullen at al, 2005, J Cer Blood Flow 25:1656-1667; Cullen et al 2006; Neurobiol Aging 27:1786-1796) . As these microhaemorrhages, are closely related to cerebral vessels, they may present triggers to activate the innate immune system, but can also indicate sites of BBB breakdown, which is considered as one of the early markers of cognitive dysfunction (Nation et al. 2019 Nat Med 25:270-276). The addition of these would enrich this Section and would avoid any perceived bias towards hypotheses that directly support PBMt, without the loss of a valid argument for the use of PBMt in conditions with early vascular dysfunction.
Thanks for this suggestion. I have added a new section “A related hypothesis concerns the influence of vascular dysfunction and micro-hemorrhages [46]. Vascular dysfunction is often described as causing vascular dementia, but there is increasing evidence that it plays a role in AD as well [47]. These micro-hemorrhages have been correlated with plaque formation [48]. These micro-hemorrhages in cerebral vessels, could act as triggers to activate the innate immune system. They could also be indicative of sites of breakdown of the blood-brain barrier, which is considered as one of the early markers of cognitive dysfunction [49]”
Minor comments:
tPBM and PBMT are not defined. PBMT only appears on the last 2 pages and it could be used interchangeably in the text, however, I assume they indicate 2 separate concepts (tPBM=transcranial PBM? PBMT=photobiomodulation treatment?) Please clarify
I have defined both “PBMT” and “tPBM” at first usage.
P3, Lines 101 & 106 Cryptochromes are misprint
corrected
P3 L105 cary misprint
corrected
P7 L 274 'uptick'?
Corrected to “a brief increase”
P8 L295. 'for' should be 'were'
corrected
P9 L372 '...it was thought...' (t is missing)
corrected
P11 L470 'deliver' misprint
corrected
P12 L478 'neurodegeneration' misprint
corrected
P12 L507 'the effects of' is repeated 2x
corrected
P19 Reference #118 is invalid citation
corrected
Reviewer 2 Report
This is a very important paper reviewing the potential roles of PBM in AD therapies. I only have a few comments that may improve the quality of writing.
Main critiques: The author used many abbreviations without definitions. It would be helpful if he can make a list of those abbreviations.
Minor critiques:
The author used PBM and tPBM (transcranial PBM?) interchangeably. Sticking to one term is needed.
Many undefined abbreviations such as rCBF (regional cerebral blood flow).
Line 507, “effects of effects of” is a typo.
Line 517, reference 147 is not Berman’s work. I guess it should be this one: Berman et al., J Neurol Neurosci. 2017; 8(1): 176.
The author forgot to write the Acknowledgments.
Author Response
This is a very important paper reviewing the potential roles of PBM in AD therapies. I only have a few comments that may improve the quality of writing.
Main critiques: The author used many abbreviations without definitions. It would be helpful if he can make a list of those abbreviations.
I do not believe that a list of abbreviations is Journal policy
Minor critiques:
The author used PBM and tPBM (transcranial PBM?) interchangeably. Sticking to one term is needed.
I have tried to use “tPBM” whenever it is clear the light is applied to the head.
Many undefined abbreviations such as rCBF (regional cerebral blood flow).
This has been defined
Line 507, “effects of effects of” is a typo.
corrected
Line 517, reference 147 is not Berman’s work. I guess it should be this one: Berman et al., J Neurol Neurosci. 2017; 8(1): 176.
The reference has been corrected
The author forgot to write the Acknowledgments.
Acknowledgments: None